# A Cyclen-Functionalized Cobalt-Substituted Sandwich-Type Tungstoarsenate with Versatility in Removal of Methylene Blue and Anti-ROS-Sensitive Tumor Cells

**DOI:** 10.3390/molecules27196451

**Published:** 2022-09-30

**Authors:** Jiai Hua, Xueman Wei, Yifeng Li, Lingzhi Li, Hui Zhang, Feng Wang, Changli Zhang, Xiang Ma

**Affiliations:** 1Chemistry and Chemical Engineering Department, Taiyuan Institute of Technology, Taiyuan 030008, China; 2Laboratory of Biochemistry and Pharmacy, Taiyuan Institute of Technology, Taiyuan 030008, China; 3Department of Geriatrics, First Affiliated Hospital of Naval Medical University, Shanghai 200081, China; 4School of Environmental Science, Nanjing Xiaozhuang University, Nanjing 211171, China; 5State Key Laboratory of Coordination Chemistry, School of Chemistry and Chemical Engineering, Nanjing University, Nanjing 210023, China

**Keywords:** sandwich-type arsenomolybdate, artificial oxidase, reactive oxygen species, methylene blue dyes, nanoscale catalyst

## Abstract

Oxidative degradation by using reactive oxygen species (ROS) is an effective method to treat pollutants. The synthesis of artificial oxidase for the degradation of dyes is a hot spot in molecular science. In this study, a nanoscale sandwich-type polyoxometalate (POM) on the basis of a tetra-nuclear cobalt cluster and trivacant B-*α*-Keggin-type tungstoarsenate {[Co(C_8_H_20_N_4_)]_4_}{Co_4_(H_2_O)_2_[HAsW_9_O_34_]_2_}∙4H_2_O (abbreviated as CAW, C_8_H_20_N_4_ = cyclen) has been synthesized and structurally examined by infrared (IR) spectrum, ultraviolet–visible (UV–Vis) spectrum, X-ray photoelectron spectrum (XPS), single-crystal X-ray diffraction (SXRD), and bond valence sum (Σ*s*) calculation. According to the structural analysis, the principal element of the CAW is derived from modifying sandwich-type polyanion {Co_4_(H_2_O)_2_ [HAsW_9_O_34_]_2_}^8^^–^ with four [Co(Cyclen)]^2+^, in which 1,4,7,10-tetraazacyclododecane (cyclen) is firstly applied to modify POM. It is also demonstrated that CAW is capable of efficiently catalyzing the production of ROS by the synergistic effects of POM fragments and Co–cyclen complexes. Moreover, CAW can interfere with the morphology and proliferation of sensitive cells by producing ROS and exhibits ability in specifically eliminating methylene blue (MB) dyes from the solution system by both adsorption and catalytic oxidation.

## 1. Introduction

Reactive oxygen species (ROS), as a biochemical medium, are ubiquitous in pathological procedures [1,2]. Hence, the development of compounds that can catalyze the production of ROS has attracted more and more attention [3,4,5]. Paired species, such as transition metal ions, are considered strong catalytic centers for constructing the reactive proteins in viable organisms, for example, hemoglobin (Hb), myoglobin (Mb), and superoxide dismutase (SOD), all of which can efficiently produce ROS [6,7,8]. In recent years, the synthesis of a range of artificial oxidase enzymes has been achieved [9,10,11]. In such materials, the activity of copper, zinc, and nickel complexes is the most potent [12]. However, less research has been focused on cobalt compounds. Furthermore, it was reported that cobalt–cyclen coordination complexes (cyclen = 1,4,7,10-tetraazacyclododecane) possess excellent ROS-catalytic ability [13]. Therefore, further research will likely lead to the construction of cobalt–cyclen complexes capable of catalyzing ROS.

Polyoxometalates (POMs), a series of compounds that feature multiform structures and useful properties, represent an attractive family of metal–oxygen clusters [14,15,16,17]. They possess several characteristics that can act as bulky polydentate ligands, such as nucleophilic oxygen-abundant surface, nanosized dimensions, and poly-bond-making sites, allowing them to bond with transition-metal ions by flexible coordination configurations [18,19,20,21,22,23,24,25,26]. Among them, the lacunary-type POMs can promote the composition of high-nuclearity clusters of transition metals and maintain them for relatively long periods [27]. Furthermore, the oxidation catalytic potential of Keggin-type building blocks has been found to potentially facilitate the catalyzation of transition-metal ion-triggered production of ROS [28,29]. A hybrid material fabricated from lacunary-type POMs partnered with high-nuclearity cobalt clusters has the potential to retain the best characteristics of both base constituents, while also exhibiting additional, unique positive characteristics.

Herein, we report a newly designed organic–inorganic hybrid Co^II^-substituted sandwich-type tungstoarsenate {[Co(C_8_H_20_N_4_)]_4_}{Co_4_(H_2_O)_2_[HAsW_9_O_34_]_2_}∙4H_2_O (abbreviated as CAW, C_8_H_20_N_4_ = cyclen), which is derived from modifying sandwich-type POM with four Co–cyclen complexes. Detailed information has been deposited at the Cambridge Crystallographic Data Centre with a CCDC number of 1944603. To our knowledge, CAW represents the first [Co(Cyclen)]^2+^-functionalized POM. As we anticipated, CAW proved to be able to efficiently catalyze the production of ROS. Moreover, it can interfere with the morphology and proliferation of PC12 cells, and specifically eliminate methylene blue (MB) dyes from the solution system by both adsorption and catalytic oxidation.

## 2. Results

### 2.1. Crystal Structure

Single-crystal structural analysis reveals that CAW consists of an electroneutral POM molecule {[Co(C_8_H_20_N_4_)]_4_}{Co_4_(H_2_O)_2_[HAsW_9_O_34_]_2_} (Figure 1A) and four lattice water. The main body of CAW (Figure 1A) is constituted by two co(cyclen)-substituted trivacant [B-*α*-AsW_9_O_34_]^9−^ fragments linked by a rhomb-like {Co_4_O_16_} group (Figure 1B), which is connected through two *μ*_4_-O atoms from two AsO_4_ tetrahedrons while the remaining four *μ*_3_-O and eight *μ*_2_-O atoms are from twelve WO_6_ moieties. As shown in Figure 1, both Co1 and Co2 cations in the sandwich belt exhibit distorted octahedral coordination environments with Co–O distances of 1.995–2.208 Å. Notably, each trivacant [B-*α*-AsW_9_O_34_]^9−^fragment was modified by two Co–cyclen complexes via *μ*_2_-O atoms from two adjacent WO_6_ octahedrons with distances of 1.998–1.999 Å. The detailed structural and anatomical schematic diagrams are shown in Figure 2.

The valence of the cobalt, tungsten, and arsenic atoms in CAW was assessed by using X-ray photoelectron spectroscopy (XPS). As shown in Figure 3A, the XPS spectrum with two peaks located at 795.1 and 779.5 eV can be assigned to Co 2*p*_1/2_ and Co 2*p*_3/2_, and the fitted curve may imply that the valences of Co in CAW are all in +2 [30,31]. As can be seen in Figure 3B, there are two peaks located at 36.9 and 34.8 eV, which could be caused by W 4*f*_7__/2_ and W 4*f*_5__/2_, respectively [32]. The fitted dashed curves in Figure 2 suggest the association of the W 4*f* peaks with W^6+^ [32]. Furthermore, as shown in Figure 3C, the fitted dashed curves at 44.28 and 39.26 eV may imply that the +5 valence As ions are present in CAW [33], which is same as the evidence from the tetrahedral configuration. 

The bond valence sums (Σ*s*) of oxygen atoms in CAW were further calculated according to the reported methods based on the bond length (Appendix A) [34]. The formula for the O atom oxidation states in CAW can be written as:$$V_{i}=\sum_{j}s_{ij}=\sum_{j}\exp\left(\frac{r_{0}\prime -r_{ij}}{B}\right)$$
where *r**_ij_* is the detected, and *r*_0_′ the theoretical, bond distances between two atoms; *B* was defined as 0.37 [35]. The theoretical values for *r*_0_′(W^6+^–O) (1.906 Å), *r*_0_′ (As^5^^+^–O) (1.767 Å), *r*_0_′ (Co^2+^–N) (1.720 Å), and *r*_0_′(Co^2+^–O) (1.692 Å ) were taken from the literature [35,36]. The calculation showed that, for the Co, W, and As in CAW, the mean valence state sum (Σ*s*) is 1.893, 5.945, and 5.186, respectively.

The results of Σ*s* by calculating using the above approach are given in Appendix A and Figure 4. POMs can easily be protonated, as their fragments are highly negatively charged, and contain abundant basic surface O atoms [34]. The 70 O atoms in CAW can be categorized into terminal O*_t_*, and bridging O*_µ_*_2_, O*_µ_*_3_, O*_µ_*_4_ types. As can be seen in Figure 4, delocalized protons are present on the O atoms with Σ*s* ranging from 0–1.50, which can thus serve as proton donors, while those with Σ*s* ranging from 1.90–2.00 hold dense electrons. The multiple protons are often described as being delocalized across the entire polyoxoanion, a frequent occurrence in POM chemistry that has been described in plenty of previous studies, for instance, [Ni(enMe)_2_]_3_[H_6_Ni_20_P_4_W_34_(OH)_4_O_136_(enMe)_8_(H_2_O)_6_]·12H_2_O [34], [Zn(en)_2_(H_2_O)_2_]_2_[(ZnO_6_)Mo_6_O_18_(As_3_O_3_)_2_] [37], and [H_3_W_12_O_40_]^5−^ [38].

As these findings suggest, delocalization of a few counter positive charges in CAW occurs in the {Co_4_(H_2_O)_2_[AsW_9_O_34_]_2_}^10−^ skeleton, which probably facilitates proton absorption and, by extension, valence state balancing upon valence alteration of the transition metal ions [39]. Therefore, it is probable that the coexistence of different POM–Co(cyclen) complexes allows the synergistic catalyzation of the production of ROS.

As can be seen in Figure 5, there are two absorption peaks in the UV spectral data of the aqueous solution, one at 195.1 nm and another broad shoulder adsorption at 190–500 nm centering on 258.5 nm. These two peaks can probably be assigned to O_t_→W and O_µ_→W charge transfer transitions, respectively [40].

The effect of pH on the stability of CAW has also been explored using UV–Vis spectroscopy. As can be seen from insets of Figure 5, insignificant variations are noted in the intensity of CAW UV–Vis absorption within a range of pH from 5.5 to 8.5 pH. Outside this range, the absorption peak intensities at 195.1 and 258.5 nm change progressively, implying the commencement of skeletal collapse within the CAW. The pH range for CAW stability can therefore be assumed to be between 5.5 and 8.5.

### 2.2. Catalytic Property

According to a recent study, transition metal ion-substituted phosphomolybdate possesses a catalytic oxidative function [9]. For this reason, dichlorofluorescein (DCF) assay was utilized to explore the CAW-mediated production of ROS. As DCF is a fluorescent probe formed through the reaction of DCFH-DA (non-fluorescent) with ROS in the presence of HRP, it can be an indicator of ROS release from the system [41]. As shown in Figure 6, CAW exhibits a much more intense fluorescent spectrum (*λ*_em_ = 528 nm) than the control group sample, suggesting a richer ROS output with CAW than without. {[C_5_H_6_N]_10_}{Co_4_(H_2_O)_2_[AsW_9_O_34_]_2_} (abbreviated as C1) and CoCl_2_ + cyclen (abbreviated as Co^2+^ + cyclen) groups also have catalytic effects. However, the CAW sample exhibits 10x more fluorescence than the C1 and 2x more than Co^2+^ + cyclen group samples. It is of note that the fluorescence intensity of the Co^2+^ group sample is very low, even lower in comparison with that of the control group, suggesting a remarkably lower output of ROS by the group with Co^2+^. This can probably be attributed to the transition metal ion–HRP enzyme interaction leading to a loss in the Co^2+^ group’s catalytic properties [42]. It is likely that, as well as facilitating synergistic production of ROS, the POM fragment in concert with the Co clusters and Co(cyclen) complexes may also avoid disturbance of the catalytic centers.

### 2.3. Cellular Oxidative Stress Injury Test

Since CAW has excellent catalytic activity in generating ROS, it may also have good inhibitory ability against ROS-sensitive tumor cells, such as neuroma cells [43]. Therefore, the inhibition of CAW toward neuronal pheochromocytoma cells (PC12) was studied by MTT assay [44]. As can be seen in Figure 7A, the cell viability of PC12 decreased with the addition of CAW gradually from 0 to 20 μM, which may imply that CAW can cause damage to PC12 cells. The concentration of 20 μM was chosen to perform the following experiment. As shown in Figure 7B, in the absence of Co–cyclen, C1 had a slight effect on the survival rate of PC12 cells compared with that of CAW. Furthermore, the addition of ascorbic acid (VC) did help alleviate the cytotoxicity of CAW. Those results may suggest that although tetra-Co-substituted sandwich-type arsenopolybdenum also has an inhibitory effect on PC12 cells to some extent, CAW mainly relies on the ROS originated from Co–cyclen complexes combined with {Co_4_(H_2_O)_2_[AsW_9_O_34_]_2_}^10−^ fragments to inhibit PC12 cells.

Further, the details of the damage of CAW to PC12 were analyzed by the cell morphological changes under the selected condition. As shown in Figure 8A, the cells in normal modality exhibit a polygonal shape, with a network of synapses that are connected in all directions. However, after incubation with CAW, as shown in Figure 8B, cell morphology atrophied, spheroidized, and synapses disappeared altogether, with widespread death. Although the viability of cells increased after the addition of VC, it can be seen from Figure 8D that it still changed greatly compared with the normal morphology, in which synapses of the cells also begin to break, turning their bodies into spheres. Therefore, we may conclude that VC only inhibits oxidative damage of CAW to some extent, but cannot completely reverse it.

### 2.4. Adsorption and Degradation of Methylene Blue (MB)

The effect of CAW on elimination of methylene blue (MB) was further evaluated by turbidity and UV–Vis spectral methods. As shown in the inset of Figure 9, the MB solution remain unchanged after 1 week at room temperature, indicating that this solution is stable under normal conditions. However, when the MB solution was coincubated with CAW, the color of the solution changed obviously, showing that CAW can make the MB solution transparent in 30 min. As shown in Figure 9, the UV–Vis spectrum of MB has four characteristic absorption peaks located at 246, 291, 611, and 665 nm, respectively [45,46]. The UV–Vis spectrum of the MB solution stored for 1 week was almost the same as that of fresh MB solution, which confirmed the above result of turbidity. When MB was incubated with CAW for 30 min, these four characteristic absorption peaks of MB almost disappeared, which is consistent with an intuitive observation. These results indicated that CAW is a powerful scavenger of MB molecules in solution.

As shown in Figure 10A–D, when observing the appearance of the CAW, after coincubation with MB, the red crystals of CAW turned into indigo blue aggregates, which may imply that the residual dyes of MB were adsorbed on the surface of CAW. As shown in Figure 10E, the IR spectrum of CAW shows similar asymmetric vibrations to other [B-*α*-AsW_9_O_34_]^9−^-containing species [47]. As can be seen in Figure 5, four characteristic spectral lines are seen at 951, 879, 753, and ~619 cm^−1^, which resulted from the *v*(As–O*_µ_*_4_), *v*(W–O*_t_*), *v*(W–O*_µ_*_2_), *v*(W–O*_µ_*_3_), and *v*(W–O*_µ_*_4_), respectively [47]. Apart from that, the characteristic spectral lines at 3258~3131 cm^−1^ can be attributable to N–H stretching vibrations [48], proving the existence of cyclen. We can ascribe the 3433 cm^−1^ vibration to stretching of the –OH bond [49]. The IR data conform to the SXRD structural elucidation. Furthermore, as shown in Figure 10F, the IR spectrum of CAW after incubation with MB is consistent with that of non-incubated CAW, which indicated that the molecular structure of CAW is maintained during the incubating process.

The presence of MB incubated with CAW in the solution was further analyzed by ESI-MS. The peaks of complexes after incubation with CAW are shown in Figure 11. As can be seen in Figure 11, no monomer or dimer peak of MB (abbreviated as compound A and the structure is shown in inset of Figure 11, cal. 284.40) or [2A + Cl^−^]^+^ peak (cal. 604.25) were found in the ESI-MS spectrum. Only one weak peak located at 589.29 was detected, which may be the complex peak of degradation product 3-((3-(dimethylamino)cyclohexa-1,4-dien-1-yl)sulfinyl)-N1,N1-dimethylbenzene-1,4-diamine (abbreviated as compound B and the structure is shown in inset of Figure 11) and MB ([A + B]^+^, cal. 589.28). The oxidation degradation process of MB was reported in the literature [50]. These results indicated that the MB molecules in the incubated solution were virtually non-existent, which is also consistent with the results of turbidity and UV–Vis spectral experiments. In summary, the results may indicate that CAW does have the ability to remove MB from the solution system by adsorption, in which the degradation of MB dyes by catalytic oxidation may be relatively weak.

## 3. Discussion

It is known that there are a lot of metalloenzymes that can produce ROS in organisms. For example, the {FeS} cluster in mitochondria can catalyze ROS by catalytic substrates of Fe^2+^/Fe^3+^ [51]. Cu ions can act as a catalytic center to yield ROS in pathologies, e.g., Cu^+^/Cu^2+^ plus Aβ protein can catalyze ROS and cause irreversible damage to neurons. The correlations between Cu^+^/Cu^2+^ and ROS generation have been well described by Prof. Kepp and Prof. Faller [1,52]. In brief, Aβ protein transformed into a misfolding conformation by Cu ions’ induction effects, in which amino acid residues coordinate with Cu ions to form unstable Cu–Aβ coordination complexes. The Cu centers can easily be reduced or oxidized in dynamic equilibrium of structural transformation, and thus catalyze the production of ROS, which can be attributed to Cu ions catalyzing Haber–Weiss system reactions [1,52,53]. Recently, it was found that trans-metal ion-substituted POMs possess a strong catalytic ability to produce ROS [54]. The adsorption of H^+^ to the POM skeleton may enhance the catalytic capacity of those complexes, since H^+^ is an important product in Haber–Weiss system reactions [53]. As shown in Figure 10, the molecular structure of CAW was stable after incubation. Hence, CAW can catalyze the generation of ROS independently and efficiently based on its special molecular structure.

The experimental results may indicate that CAW was involved in the solution precipitation equilibrium. Furthermore, the images of CAW incubated with MB may show that MB and its fragments were adsorbed on the CAW surface. These results may suggest that CAW should eliminate MB by adsorption as the first step, which seems to be similar to most POMs that can adsorb MB by electrostatic or π–π stacking effects [55]. However, ESI-MS analysis showed that there were some MB degradation fragments in the solution system, which may imply that some MB molecules are not only adsorbed on CAW, but also degraded by oxidation. More importantly, although it may not be identified as a strong oxidation, the oxidation of CAW can be carried out under mild conditions, without light or heat.

## 4. Materials and Methods

### 4.1. Materials

All reagents applied in the current work were analytically pure and employed as received. 2′,7′-dichlorofluorescein diacetate (DCFH-DA) was obtained from Sigma-Aldrich, while CoCl_2_·2H_2_O, Na_2_WO_4_·2H_2_O, NaAsO_2_, and cyclen (1,4,7,10-tetraazacyclododecane) were obtained from J *&* K Scientific. 3-(4,5-dimethyl-2-thiazolyl)-2,5-diphenyl-2-H-tetrazolium bromide (MTT), nerve growth factor 7S (NGF-7S), and tris(hydromethyl)aminomethane (Tris) were purchased from Sigma-Aldrich Inc. (USA). Pheochromocytoma cells (PC12 cells) were purchased from American Type Culture Collection (ATCC). Milli-Q water (Merck) was used to prepare all of the solutions, and a Millipore filter (0.22 µm) was used for all filtrations.

Spectral analysis of the CAW samples was performed using a Nicolet 170 SX FTIR spectrometer scanning from 4000~400 cm^−1^, a PHI5000 VersaProbe X-ray Photoelectron Spectrometer, and a Thermo Scientific Varioskan Flash Microplate Reader (for measuring DCF fluorescence).

### 4.2. Synthesis of CAW

First, we prepared two solutions. Solution A consisted of 9.90 g of Na_2_WO_4_·2H_2_O (30.00 mmol) and 1.80 g of NaAsO_2_ (13.40 mmol) in 100 mL of water. Solution B consisted of 1.80 g of CoCl_2_·2H_2_O (10.50 mmol) and 1.70 g cyclen (10.00 mmol) in 100 mL of water. Both samples were prepared using mechanical stirring. After their preparation, the two solutions were combined. The mixture was stirred for 10 min at ambient temperature. Subsequently, 6 mol·L^−1^ HCl was dripped dropwise into the mixture to shift the pH to 5.8. The mixture was then stored for three days at 160 °C in a Teflon-lined autoclave (25 mL) before being cooled to ambient temperature. Afterwards, the CAW red crystals were harvested from the solution by filtration. The yield was found to be 29%.

### 4.3. X-ray Data Collection and Structure Refinement

A single CAW crystal was mounted in an Apex-2 diffractometer (Bruker) with a CCD detector using graphite monochromatized Mo K*α* radiation (*λ* = 0.71073 Å) at 296 K. The SAINT software package (Bruker) was used for data integration [56]. Lorentz and polarization corrections were made in the standard way. Adsorption corrections were made using the multiscan approach with the aid of the SADABS software package (Bruker) [57]. After being solved directly, we refined the structure with the full-matrix least-squares procedure on *F*^2^. This same refinement was performed successively along with Fourier syntheses for the remaining atoms. The SHELXL-97 package (Georg-August-Universität Göttingen) was used for all computations [58]. The Fourier difference map did not show the positions of any of the water molecule-related hydrogen atoms. Geometrical positioning was used for hydrogen atoms attached to C and N atoms. Isotropic refinement was carried out for all the hydrogen atoms using the riding model, using the default variables within SHELXL. A summary of crystal data and structure refinements for CAW (CCDC No. 1944603) is listed in Table 1. The selected bond length of CAW and experimental methods are summarized in the Electronic Supplementary Information (ESI).

### 4.4. Catalytic ROS Production of CAW

A 1 mM stock solution of DCFH-DA was prepared using Tris-buffer (20 mM Tris–HCl/150 mM NaCl pH 7.4) and the procedure described in reference [41]. The same buffer was used to prepare a 4 µM horseradish peroxidase (HRP) stock solution. All samples were incubated at ambient temperature after adding 10 µM ascorbate that either did or did not contain CAW (0.025 mM), and then 200 µL of each solution was pipetted into one well of a black 96-well flat-bottomed microplate. DCFH-DA (100 µM) and HRP (0.04 µM) were supplemented, and then the samples were left in the dark at ambient temperature for an additional 12 hours. Fluorescent spectra were captured over a range from 505nm to 650 nm with a microplate reader (Varioskan Flash, Thermo Scientific) with *λ*_ex_ = 485 nm. Spectra of {[C_5_H_6_N]_10_}{Co_4_(H_2_O)_2_[AsW_9_O_34_]_2_} (C1, 0.025 mM), CoCl_2_ (Co^2+^, 0.200 mM), CoCl_2_ + cyclen (Co^2+^ + cyclen, 0.200 mM), and the control group were also captured for comparison under the same conditions as presented above.

### 4.5. Cellular Oxidative Stress Injury Test

PC12 cells were cultured by a method from the literature [59]. The mature PC12 cells were incubated with CAW at different concentration gradients from 0 to 20 μM for 24 h. Ascorbic acid and CAW (20 μM) were incubated together as the control group. The control group was used for comparison under the same conditions as presented above. The MTT kit was then used for testing. Cell morphology was observed under an inverted fluorescence microscope.

### 4.6. Adsorption and Degradation of Methylene Blue (MB)

The CAW (250 mg) was mixed with methylene blue solution (160 μM) and stirred in a darkroom for 30 min. Then, the solids were separated from the supernatant by centrifugation (5000 rpm). The UV–Vis spectra of supernatant were recorded on a UV-3600 spectrometer from 200 to 900 nm. The IR spectra of CAW before and after incubation with MB were recorded on a NICOLET iS10 spectrometer in the range of 400–4000 cm^−1^. The ESI-MS spectra of the supernatant were measured by LCQ Fleet mass spectrometer.

## 5. Conclusions

To conclude, a Co–cyclen functionalized nanoscale tetra-nuclear cobalt cluster sandwiched tungstoarsenate {[Co(C_8_H_20_N_4_)]_4_}{Co_4_(H_2_O)_2_[HAsW_9_O_34_]_2_}∙4H_2_O (abbreviated as CAW, C_8_H_20_N_4_ = cyclen) has been designed and synthesized successfully. CAW was constituted by two Co(cyclen)-substituted trivacant [B-*α*-AsW_9_O_34_]^9−^ fragments linked by a rhomb-like {Co_4_O_16_} group forming a sandwich-type structure. It is effective in catalyzing ROS generation, which can probably be attributed to the synergic interplay between POM fragments and Co–cyclen complexes, and further inhibits the ROS-sensitive PC12 cells. Furthermore, CAW can specifically eliminate methylene blue (MB) dyes from the solution system by both adsorption and catalytic oxidation effects. CAW is likely to have a broad range of possible applications in inorganic chemistry and biochemistry studies due to its neoteric structure and useful properties.

## Figures and Tables

**Figure 1 molecules-27-06451-f001:**
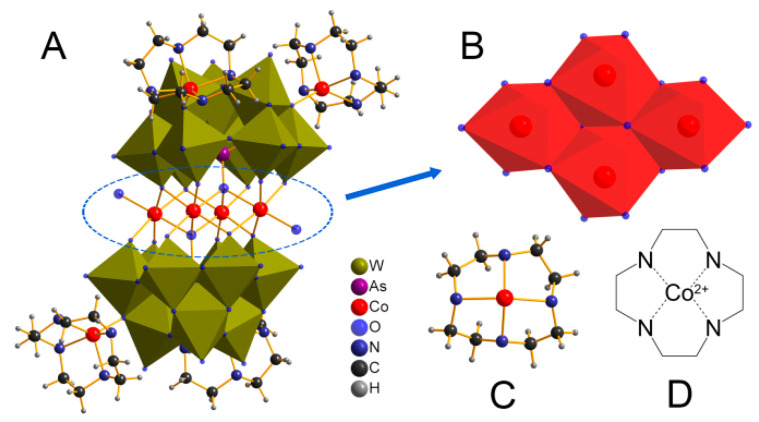
Schematic diagram of the single crystal structure of {[Co(C_8_H_20_N_4_)]_4_}{Co_4_(H_2_O)_2_ [HAsW_9_O_34_]_2_} (CAW) and its components. (**A**) Ball-and-stick view/polyhedral view of the CAW; (**B**) polyhedral representation of tetra-Co cluster {Co_4_(H_2_O)_2_O_14_}; ball-and-stick view and structural diagram of [Co(Cyclen)]^2+^ for (**C**,**D**).

**Figure 2 molecules-27-06451-f002:**
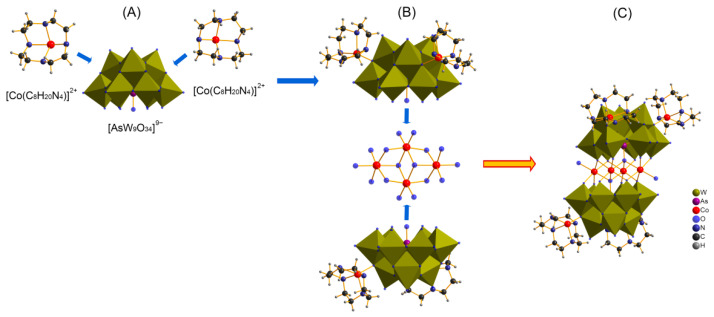
The anatomical view of the {[Co(C_8_H_20_N_4_)]_4_}{Co_4_(H_2_O)_2_[HAsW_9_O_34_]_2_} (CAW). (**A**) The decomposition of {[Co(C_8_H_20_N_4_)]_2_[AsW_9_O_34_]}^5−^ into two [Co(C_8_H_20_N_4_)]^2+^ and one [AsW_9_O_34_]^9−^; (**B**) the decomposition of {[Co(C_8_H_20_N_4_)]_4_}{Co_4_(H_2_O)_2_[HAsW_9_O_34_]_2_} into two {[Co(C_8_H_20_N_4_)]_2_[AsW_9_O_34_]}^5−^ and one {Co_4_(H_2_O)_2_O_14_} cluster; (**C**) the single crystal structure of {[Co(C_8_H_20_N_4_)]_4_}{Co_4_(H_2_O)_2_ [HAsW_9_O_34_]_2_}.

**Figure 3 molecules-27-06451-f003:**
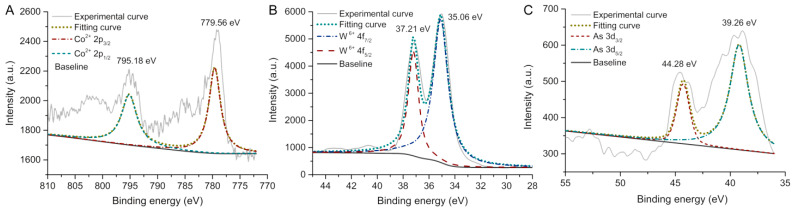
X-ray photoelectron spectrum (XPS) and the fitted curves of (**A**) Co (with 2p_3/2_ 779.56 eV and 2p_1/2_ 795.18 eV), (**B**) W (with 4f_7/2_ 35.06 eV and 4f_5/2_ 37.21 eV), and (**C**) As (with 3d_5/2_ 39.26 eV and 3d_3/2_ 44.28 eV) in {[Co(C_8_H_20_N_4_)]_4_}{Co_4_(H_2_O)_2_ [HAsW_9_O_34_]_2_}.

**Figure 4 molecules-27-06451-f004:**
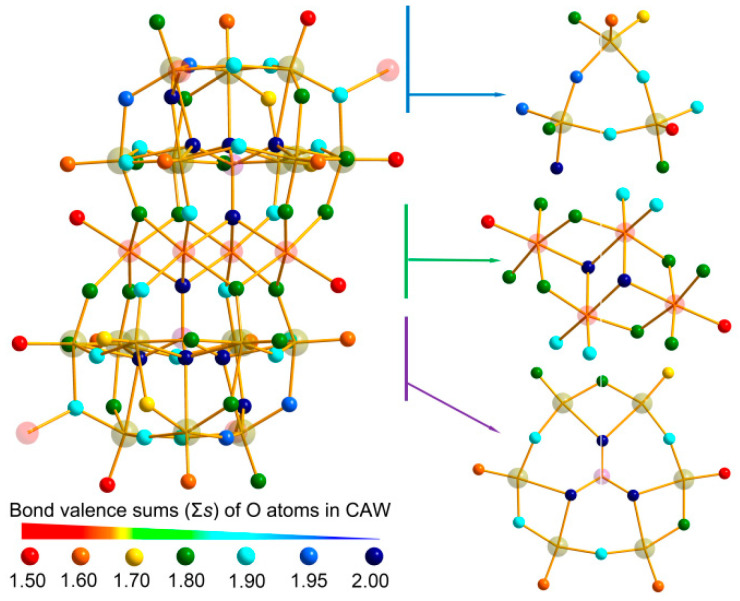
Σ*s* of O atoms in the polyanion of {[Co(C_8_H_20_N_4_)]_4_}{Co_4_(H_2_O)_2_[HAsW_9_O_34_]_2_}, the Σ*s* values for each O atom are represented by the color scheme shown.

**Figure 5 molecules-27-06451-f005:**
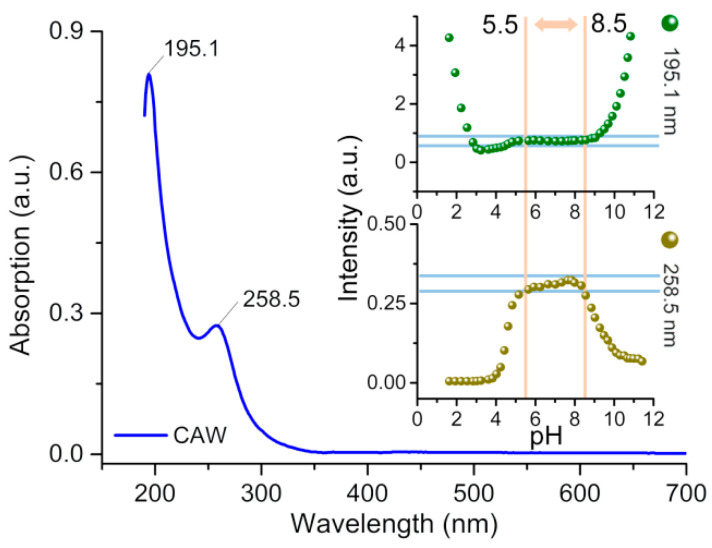
The UV–Vis spectrum of CAW in ultrapure water. Also shown: the variation in the peak intensities with pH of the solution (inset figures).

**Figure 6 molecules-27-06451-f006:**
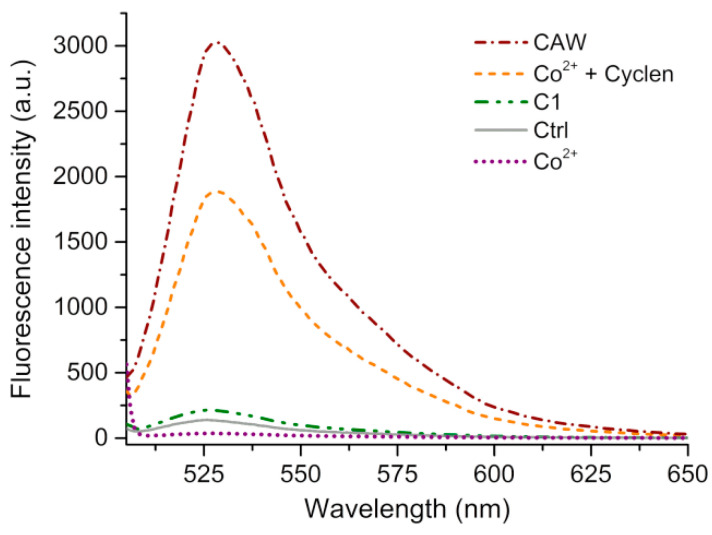
The intensity of the fluorescence of DCF (*λ*_ex_ = 485 nm, *λ*_em_ = 650 nm) in pH 7.4 Tris-buffer (20 mM Tris-HCl/150 mM NaCl) induced by {[Co(C_8_H_20_N_4_)]_4_}{Co_4_(H_2_O)_2_ [HAsW_9_O_34_]_2_} (CAW, 0.025 mM), {[C_5_H_6_N]_10_}{Co_4_(H_2_O)_2_[AsW_9_O_34_]_2_} (C1, 0.025 mM), CoCl_2_ (Co^2+^, 0.200 mM), CoCl_2_ + cyclen (Co^2+^ + cyclen, 0.200 mM), and the control group samples at 37 °C.

**Figure 7 molecules-27-06451-f007:**
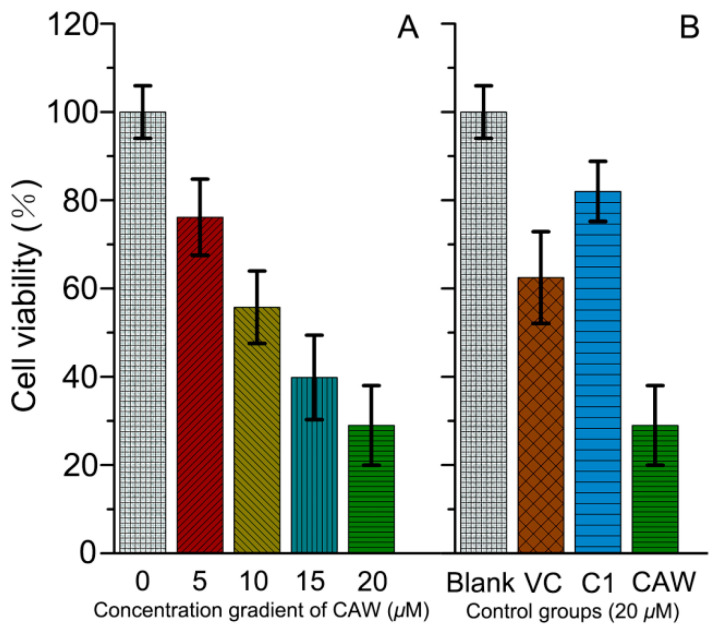
PC12 cell viability determined by MTT assay. (**A**) After incubation with CAW of different concentration gradients (0, 5, 10, 15, 20 μM) for 24 h. (**B**) Incubation of different control groups (Ctrl: blank group; VC: 20 μM CAW + 20 μM ascorbic acid; CAW: 20 μM CAW; C1: 20 μM {[C_5_H_6_N]_10_}{Co_4_(H_2_O)_2_[AsW_9_O_34_]_2_}) at 37 °C for 24 h.

**Figure 8 molecules-27-06451-f008:**
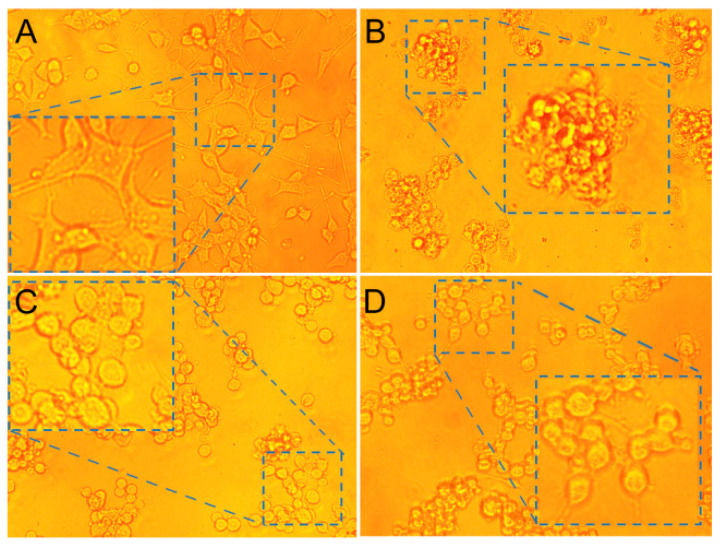
Photomicrographs of cells after incubation with 20 μM CAW for (**B**), 20 μM {[C_5_H_6_N]_10_}{Co_4_(H_2_O)_2_[AsW_9_O_34_]_2_} for (**C**), 20 μM CAW + 20 μM ascorbic acid for (**D**), and blank group for (**A**) in the culture medium at 37 °C for 24 h.

**Figure 9 molecules-27-06451-f009:**
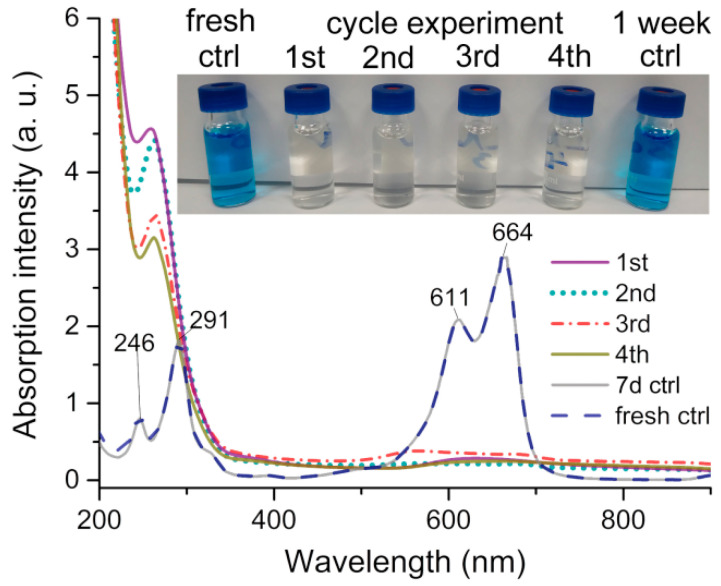
Adsorption and degradation effect of CAW (250 mg) on the methylene blue solution (MB, 160 μM) determined by turbidimetry (inset figure) and UV–Vis spectra in the deionized water at R.T.

**Figure 10 molecules-27-06451-f010:**
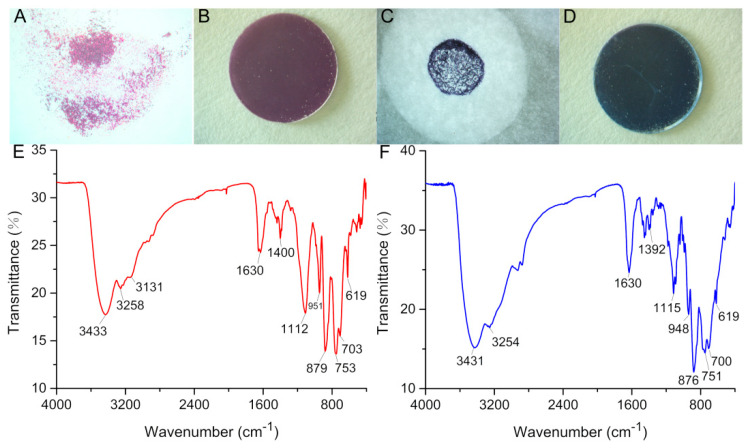
Crystal micrograph for (**A**), color of KBr tablets for (**B**), and IR spectrum for (**E**) of unincubated CAW; and crystal micrograph for (**C**), color of KBr tablets for (**D**), and IR spectrum for (**F**) of CAW incubated with MB.

**Figure 11 molecules-27-06451-f011:**
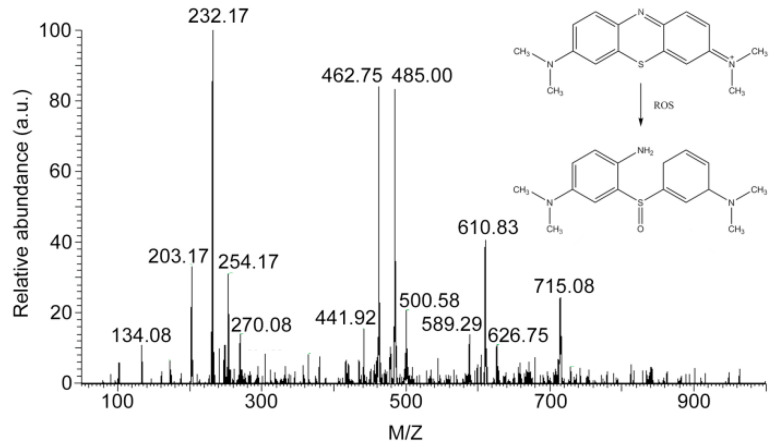
ESI-MS spectrum of MB incubated with CAW and the oxidative degradation process of methylene blue (MB) according to the literature (inset of figure, [50]).

**Table 1 molecules-27-06451-t001:** Crystallographic data and structural refinements for CAW.

Empirical formula	C_32_H_64_As_2_Co_8_N_16_O_74_W_18_
Formula weight	5787.57
Crystal system	Monoclinic
Space group	*C*2/*c*
*a*/Å	24.2054 (15)
*b*/Å	15.6298 (10)
*c*/Å	28.7759 (18)
*α*/deg	90
*β*/deg	97.3715(9)
*γ*/deg	90
*V*/Å^3^	10796.7 (12)
*Z*	4
*D*_c_/g cm^−3^	3.561
*μ*/mm^−1^	20.986
*T*/K	296.15
Limiting indices	–28 ≤ h ≤ 28,
	–18 ≤ k ≤ 18,
	–34 ≤ l ≤ 34
Measured reflections	52408
Independent reflections	9613
*R* _int_	0.1132
Data/restraints/parameters	9613/90/666
GOF on *F*^2^	1.029
Final *R* indices (*I* > 2*σ*(*I*))	*R*_1_ = 0.0467,
	*wR*_2_ = 0.0905
*R* indices (all data)	*R*_1_ = 0.0877
	*wR*_2_ = 0.1071
Completeness	99.90%

## Data Availability

Crystallographic data for the structural analysis have been deposited with the Cambridge Crystallographic Data Centre, CCDC reference number: 1944603 for CAW. These data can be obtained free of charge from the Cambridge Crystallographic Data Centre via www.ccdc.cam.ac.uk/data_request/cif.

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
