# Peer review of "A Cyclen-Functionalized Cobalt-Substituted Sandwich-Type Tungstoarsenate with Versatility in Removal of Methylene Blue and Anti-ROS-Sensitive Tumor Cells"

_molecules, 2022, doi:10.3390/molecules27196451_

Round 1
Reviewer 1 Report
The manuscript describes a novel artificial oxidase mimics capable of producing reactive oxygen species. This inorganic/organic polynuclear metal complex was synthesized and characterized structurally, spectroscopically and functionally. I have several remarks, mostly minor, that should be addressed before acceptance.
Based on section numbers, one section (Discussion?) is misssing, Note that section 5. Conclusions follows section 3. Materials and Methods.
Figure 1 caption does not contain a caption for the whole figure. Most figure captions are too short to allow understanding the figures without reference to the text.
The authors should check that detailed conditions (medium, temperature, etc.) are provided for each experiment to allow its repetition by the reader.
Careful language editing is required. The title of the paper does not make sense in the current form. Other examples are, but not limited to lines 115, 223.
Author Response
Reviewer: #1
Thank you for your positive comments. We have revised our manuscript according to the suggestions.
Comment (C) 1: Based on section numbers, one section (Discussion?) is misssing, Note that section 5. Conclusions follows section 3. Materials and Methods.
Answer (A) 1: Thanks a lot. The Discussion section has been inset into the manuscript as section 4.
C 2: Figure 1 caption does not contain a caption for the whole figure. Most figure captions are too short to allow understanding the figures without reference to the text.
A 2: Thanks for your suggestion. The captions of Fig.1-10 have been revised to add relevant information.
C3: The authors should check that detailed conditions (medium, temperature, etc.) are provided for each experiment to allow its repetition by the reader.
A3: Thank you for your suggestion. The details of the experiment have been improved.
C4: Careful language editing is required. The title of the paper does not make sense in the current form. Other examples are, but not limited to lines 115, 223.
A4: Thank you for your suggestion. The title has been renamed as “A Cyclen-functionalized Cobalt substituted sandwich-type tungstoarsenate with versatility in removal of methylene blue and anti-ROS sensitive tumor cells”, and the language including lines 115 and 223 have also been revised and improved.

Reviewer 2 Report
This manuscript by Jiai Hua et al represents a very intersting and scientifically sound work, in which a nanoscale sandwich-type polyoxometalate (POM) on the basis of a tetra-nuclear cobalt cluster and trivacant B-α-Keggin-type tungstoarsenate has been synthesized and structurally featured by several spectroscopic techniques. Synthesized CAW showed to be effective in interfere with the morphology and proliferation of sensitive cells by producing ROS; and exhibits ability in specifically eliminating methylene blue (MB) dyes from the solution system by both adsorption and catalytic oxidation. Therefore, I strongly recommend the publication of this manuscript in the present form.
Author Response
Reviewer: #2
Thank you for your careful reviews and positive comments.

Reviewer 3 Report
Manuscript: molecules-1871576
Title: A heterogeneous catalyzed oxidase consists of [Co(Cyclen)]2+ 3 functionalized sandwich-type tungstoarsenate with ROS 4 catalytic activity
In this manuscript, a nanoscale sandwich-type polyoxometalate (POM) on the basis of a tetra-nuclear cobalt cluster and trivacant B-α-Keggin-type tungstoarsenate {[Co(C8H20N4)]4}{Co4(H2O)2[HAsW9O34]2}∙4H2O (abbreviated as CAW, C8H20N4 = Cyclen) has been synthesized for oxidative degradation of pollutants by reactive oxygen species (ROS). The 1,4,7,10-tetraazacyclododecane (Cyclen) is firstly applied to modify POM, and this designed material is well characterized. This topic can be suitable for publication in this journal but a major revision is still needed for further improvement. The comments are listed as follows:
1. There is As element in the as-designed organic-inorganic hybrid CoII-substituted sandwich-type tungstoarsenate, so CAW may be not an environmental friendly materials for practical application. Please prove it stability during the oxidative degradation process of pollutants.
2. According to the XPS spectrum in Fig 3a, the valence of the cobalt can be mixed 2+ and 3+. Please refer to the literature: Applied Catalysis B: Environmental 210 (2017) 454-461; Nano Energy 89 (2021) 106326.
3. The formation of ROS species (such as·O2,·OH, etc) can be further confirmed by ESR analysis.
The removal of MB dyes through adsorption and ROS degradation, herein can the contribution of degradation be greater than adsorption in this process? It should be further clarified.
4. As for the characteristic absorption peaks of MB dye located at 246, 291, 611, and 665 nm in the UV-vis spectrum, suitable references should be added, such as Catal. Sci. Technol., 2018,8, 6180-6195, Catal. Sci. Technol., 2021, 11, 4181-4195.
5. The figure 4 and table 1 can be shown in supporting information instead of the main text.
6. From the ESI-MS spectrum of MB incubated by CAW, it can be concluded that the degradation of MB dyes may be relatively weak and the CAW catalyst may be not stable.
7. How about the ability of ROS generation under light irradiation for the CAW catalyst? The oxidative degradation process may be more efficient.
Author Response
Reviewer: #3
Thank you for your positive comments. We have revised our manuscript according to the suggestions.
Comment (C) 1: There is As element in the as-designed organic-inorganic hybrid CoII-substituted sandwich-type tungstoarsenate, so CAW may be not an environmental friendly materials for practical application. Please prove it stability during the oxidative degradation process of pollutants.
Answer (A) 1: Thanks a lot for your suggestion. The Structural stability experiments of CAW have been added to the text.
C2: According to the XPS spectrum in Fig 3a, the valence of the cobalt can be mixed 2+ and 3+. Please refer to the literature: Applied Catalysis B: Environmental 210 (2017) 454-461; Nano Energy 89 (2021) 106326.
A2: Thanks for your suggestion. The two papers have been cited in the manuscript.
C3: The formation of ROS species (such as·O2,·OH, etc) can be further confirmed by ESR analysis. The removal of MB dyes through adsorption and ROS degradation, herein can the contribution of degradation be greater than adsorption in this process? It should be further clarified.
A3: Thank you for your suggestion. After experimental verification, we agree with your opinion. The contribution of adsorption is greater than degradation in the process. We have revised the relevant statements in the text to highlight that the main effect of CAW on MB removal is adsorption performance. We have tested it by using ESR analysis, and there are many unexplained species. Since the focus of this paper is not to clarify the specific species of ROS catalyzed by CAW, we do not include this part in the main text.
C4: As for the characteristic absorption peaks of MB dye located at 246, 291, 611, and 665 nm in the UV-vis spectrum, suitable references should be added, such as Catal. Sci. Technol., 2018,8, 6180-6195, Catal. Sci. Technol., 2021, 11, 4181-4195.
A4: Thank you for your suggestion. The literature have been cited in the text.
C5: The figure 4 and table 1 can be shown in supporting information instead of the main text.
A5: Thank you for your suggestion. The Table 1 has been move into the ESI as Table S1. Figure 4 and Table 1 are different representations of the BVS calculation. Since Table 1 has been moved into the ESI, we believe that figure 4 should be better retained in the text.
C6: From the ESI-MS spectrum of MB incubated by CAW, it can be concluded that the degradation of MB dyes may be relatively weak and the CAW catalyst may be not stable.
A6: We agree with you. In summary, the results may indicate that CAW does have the ability to remove MB from the solution system by adsorption, in which the degradation of MB dyes by catalytic oxidation may be relatively weak.
C7: How about the ability of ROS generation under light irradiation for the CAW catalyst? The oxidative degradation process may be more efficient.
A7: Thanks for your suggestion. We irradiated the MB removal experiment by using a UV lamp in a dark room. The results showed that the oxidation capacity of CAW was not improved greatly.

Round 2
Reviewer 1 Report
The authors have substantially improved the manuscript.
As to the title, "ROS-sensitive" may be more appropriate than "anti-ROS sensitive".
Reviewer 3 Report
The previous comments have been well addressed and the revised manuscript can be considered for acceptance.